

# Long noncoding RNA MEG3 suppresses podocyte injury in diabetic nephropathy by inactivating Wnt/β-catenin signaling

Xiajing Che[1,*], Xin Deng[2,*], Kewei Xie[1], Qin Wang[1], Jiayi Yan[1], Xinghua Shao[1], Zhaohui Ni[1] and Liang Ying[3]

[1] Department of Nephrology, RenJi Hospital, School of Medicine, Shanghai JiaoTong University, Shanghai, China
[2] Department of Nephrology, Changshu NO. 1 People Hospital, Jiangsu, China
[3] Department of Urology, RenJi Hospital, School of Medicine, Shanghai JiaoTong University, Shanghai, China
* These authors contributed equally to this work.

Corresponding authors
Zhaohui Ni, nizhaohui@renji.com
Liang Ying, yingliang@renji.com

## ABSTRACT

**Background:** Diabetic nephropathy (DN) is one of the principal complications of diabetes and podocyte injury plays an important role in the DN pathogenesis. Wnt/β-catenin signaling overactivation confers podocyte injury and promotes multiple types of renal disease. However, the underlying mechanism of Wnt/β-catenin signaling activation in DN progression has not been fully elucidated. Long noncoding RNA (lncRNA) is a large class of endogenous RNA molecules lacking functional code capacity and which participates in the pathogenesis of human disease, including DN.

**Method:** A diabetes model was constructed by intraperitoneal injection of Streptozotocin in rats. The MPC5 cells were used to create the in vitro model. Western blot and Quantitative reverse-transcriptase-PCR were used to examine the expression of protein and mRNA. The migrated capacity was analyzed by Transwell migration assay. The cell viability was detected by CCK8.

**Results:** In the present study, we revealed the association of lncRNA Maternally Expressed Gene 3 (MEG3) with aberrant activation of Wnt/β-catenin signaling and the role of MEG3/Wnt axis in podocyte injury. We found that high glucose (HG) treatment suppressed MEG3 expression in cultured podocytes, activated Wnt/β-catenin signaling and caused podocyte injury as indicated by the downregulation of podocyte-specific markers (podocin and synaptopodin) and the upregulation of snail1 and α-smooth muscle actin. Overexpression of MEG3 attenuated HG-induced podocyte injury by reducing Wnt/β-catenin activity, repressing cell migration, reactive oxygen species production and increasing the viability of podocytes. Furthermore, we provided evidences that restoration of Wnt/β-catenin signaling by specific agonist impeded the protective effect of MEG3 on podocyte injury. Current results demonstrated that MEG3/Wnt axis plays an important role in fostering podocyte injury and may serve as a potential therapeutic target for the treatment of DN.

**Conclusion:** lncRNA MEG3 ameliorates podocyte injury in DN via inactivating Wnt/β-catenin signaling.

## INTRODUCTION

Diabetic nephropathy (DN), the primary cause of end-stage kidney diseases, is a major complication of diabetes mellitus (DM) (*John, 2016*; *Schena & Gesualdo, 2005*; *Van Buren & Toto, 2012*). Podocytes are a crucial constituent of renal filtration barrier and their injury is a primary event in the development of various glomerular diseases such as DN (*Nagata, 2016*; *Shankland, 2006*). The podocyte number is a balance between podocyte injury (e.g., death, apoptosis and detachment) and proliferation (*Shankland, 2006*). Reduced podocyte number results in glomerulosclerosis and proteinuria in nondiabetic and diabetic glomerular diseases (*Zhou et al., 2017*). The causes of podocyte injury are multiple.

It is well-known that the Wnt/β-catenin signaling plays an important role in homeostasis, organogenesis and human diseases (*Greka & Mundel, 2012*). In patients with diabetes and streptozotocin (STZ)-induced diabetic mice, the activation of Wnt/β-catenin signaling is upregulated in podocytes of DN (*Bose, Almas & Prabhakar, 2017*; *Maezawa, Takemoto & Yokote, 2015*) and overactivation of Wnt/β-catenin signaling results in podocyte injury (*Dai et al., 2009*). The expression of Wnt proteins (e.g., Wnt1, Wnt4 and Wnt6) is also increased in STZ-induced diabetic mice (*Bose, Almas & Prabhakar, 2017*). More important, inhibition of Wnt/β-catenin signaling partially recovers podocyte function (*Bose, Almas & Prabhakar, 2017*).

Long noncoding RNAs (lncRNAs) are a class of non-protein coding transcripts (more than 200 nt) which have been demonstrated widely as vital regulators in gene expression, cellular function and disease processes (*Boon et al., 2016*). Mounting evidence have demonstrated that lncRNAs play an important role in the progression of DN (*Alvarez & Distefano, 2013*). For example, lncRNA MALAT1 expression is increased in experimental DN and MALAT1 inhibition attenuates high glucose (HG)-induced podocyte injury by regulating Wnt/β-catenin signaling (*Gutschner, Hämmerle & Diederichs, 2013*; *Hu et al., 2017*). The roles of other lncRNAs (e.g., Tug1, Erbb4-IR, Gm4419, TCF7, etc.) in DN have also been identified (*Liu & Sun, 2019*; *Long et al., 2016*; *Sun et al., 2018*; *Yi et al., 2017*). Our previous study showed that lncRNA Maternally Expressed Gene 3 (MEG3) is downregulated in bladder cancer and regulates cancer cell proliferation by inhibiting Wnt/β-catenin signaling and autophagy (*Ying et al., 2013*). Given the important role of Wnt/β-catenin signaling in the progression of DN, we speculated that MEG3 might be associated with DN. Additionally, MEG3 was recently reported to be involved in DM-related microvascular dysfunction (*Qiu et al., 2016*). Genome imprinting study showed that MEG3 gene region on chromosome 14q32.2 is associated with the susceptibility of type one diabetes (*Wallace et al., 2010*). Therefore, the present study investigated the role of MEG3 in podocyte injury and DN and our data showed that MEG3 attenuated HG-induced podocyte injury by repressing Wnt/β-catenin activity.

## MATERIALS AND METHODS

### Animal model

All animal experiments were undertaken at Shanghai Jiaotong University School of Medicine and approved by the Animal Ethics Committee of Shanghai Jiaotong University School of Medicine (SYKX-2008-0050). Male Sprague Dawley rats (6–8 weeks of age) were purchased from the SLAC Laboratory Animal Co. Ltd. (Shanghai, China) and kept in a specific-pathogen-free environment. Diabetes were induced by intraperitoneal injection of STZ (70 mg/kg B.W. in a 0.1 M citrate buffer) (Sigma–Aldrich, St. Louis, MO, USA) along with only citrate buffer in the control. The blood glucose level was then examined until it is above 16.7 mmol/L as hyperglycemia was determined.

### Cell culture and MEG3 overexpression

MPC5, a murine podocyte cell line from American Type Culture Collection (ATCC, Manassas, VA, USA), was cultured in Dulbecco's Modified Eagle's Medium (DMEM, Gibco, CA, USA) supplemented with 10% FBS (Gibco). Cells were maintained in a humidified incubator at 37 °C with 5% $CO_2$. In order to carry out MEG3 overexpression, plasmid pcDNA3-MEG3 was constructed by introducing the full-length MEG3 into the pcDNA3.1 as our previously described (*Ying et al., 2013*). pcDNA3-MEG3 was then transfected into MPC5 cells using Lipofectamine 2000™ reagent (Invitrogen, Carlsbad, CA, USA) according to the manufacturer's instructions.

The TurboFect in vivo transfection reagent (Thermo Scientific, Wilmington, NC, USA) was used as an in vivo delivery vehicle for pcDNA3-MEG3 to prevent degradation and enhance transfection efficiency as described previously (*Mao et al., 2019*; *Xu et al., 2014*).

### Quantitative reverse-transcriptase-PCR

The RNA expression level of lncRNA and mRNA was assessed using Quantitative reverse-transcriptase-PCR (qPCR). Total RNA of cells or tissues was extracted from using Trizol reagent (Invitrogen, Carlsbad, CA, USA). The high capacity cDNA Reverse Transcription Kit (Invitrogen, Carlsbad, CA, USA) was used for reverse transcription. The SYBR™ Green PCR Master Mix (Applied Biosystems, Foster City, CA, USA) was used for qPCR amplification on StepOne Plus real-time PCR system (Thermo Fisher, Waltham, MA, USA). All qPCRs were carried out in triplicate for each sample and β-actin was used as endogenous control for mRNAs and lncRNA.

### Cell migration assay

The migrated capacity of MPC5 cells was analyzed using Transwell migration assay. For transwell migration assay, the 24-well plate transwell system (8 μm pore size; Corning–Costar, Corning, NY, USA) were used. The filter in the upper chamber was incubated for 2 h at 37 °C. Then, MPC5 cells were plates (100,000 cells/well) in the upper chamber with serum-free DMEM media and DMEM media with 10% FBS was added to the bottom chamber. Cells were incubated at 37 °C with 5% $CO_2$ and allowed to migrate for 24 h. The cells that had migrated were fixed with 4% formaldehyde and stained with

0.1% crystal violet and captured with a microscope (LEICA, Wetzlar, Germany). The experiments were performed in triple set and results were obtained at four randomized visual fields.

## Immunohistochemistry

The protein expression of β-catenin, snail1 and synaptopodin in tissues was assayed using Immunohistochemistry (IHC). Renal tissues were fixed with 10% formalin and embedded in paraffin and the embedded samples were cut into 5 μm sections and mounted on slides. After antigen retrieval, the sections were incubated with primary antibodies for anti-β-catenin (1:500, ab32572; Abcam, CA, USA) or anti-snail1 (1:500, ab53519) or anti-synaptopodin (1:500, ab117702) at 4 °C overnight. After washing three times with PBS, the sections were then incubated with the secondary antibody (HRP labeled goat anti rabbit, 1:200; CST, CA, USA). β-catenin, snail1 and synaptopodin expression were visualized using DAB staining.

## Western blot analysis

The protein expression of β-catenin, snail1, a-SMA and podocin in cells or tissues was assayed using western blot analysis. Total protein of MPC5 cells or tissues was extracted in RIPA lysis buffer and then quantified by BCA assay (Thermo Fisher Scientific, Waltham, MA, USA). A total of 50 μg protein was separated by 10% SDS-PAGE and then electro-transferred to polyvinylidene difloride membrane. The membrane was then blocked with TBST containing 5% nonfat dry milk at 4 °C overnight. β-catenin (1:1000, ab32572; Abcam, CA, USA), snail1 (1:1000, ab53519), α-smooth muscle actin (α-SMA) (1:1000, ab5831), and podocin (1:1000, ab50339) were used as the primary antibodies. After washing three times with TBST, the membrane was incubated with appropriate secondary antibodies (1:1000; Abcam, CA, USA). Protein bands were visualized using the ECL chemiluminescence system (BioRad, Hercules, CA, USA). β-actin was set as the internal control.

## Cell viability

Cell Counting Kit-8 (CCK-8; Beyotime Institute of Biotechnology, Jiangsu, China) was carried out to determine the viability of MPC5 cells. In brief, MPC5 cells transfected with pcDNA3-MEG3 ($3 \times 10^3$ cells/well) were plated into 96-well plates and cultured. A total of 20 μL of CCK-8 liquid was added at the indicated time. Following incubation for 2 h at 37 °C, the optical density was detected at 450 nm through a microplate reader (Bio-Rad Model 550).

## Reactive oxygen species production

Intracellular reactive oxygen species (ROS) generation in MPC5 cells was assayed using ELISA kit (Vazyme Biotech, Nanjing, China) according to the manufacturer's protocol. In brief, cells were plated in a 96-well plate and cultured with their respective treatments for 24 h. Culture medium was removed and plates were washed thrice at 24 h following culture. Fluorescence was measured by a Victor3 1420 Multilabel Counter (PerkinElmer Instruments, Shanghai, China).

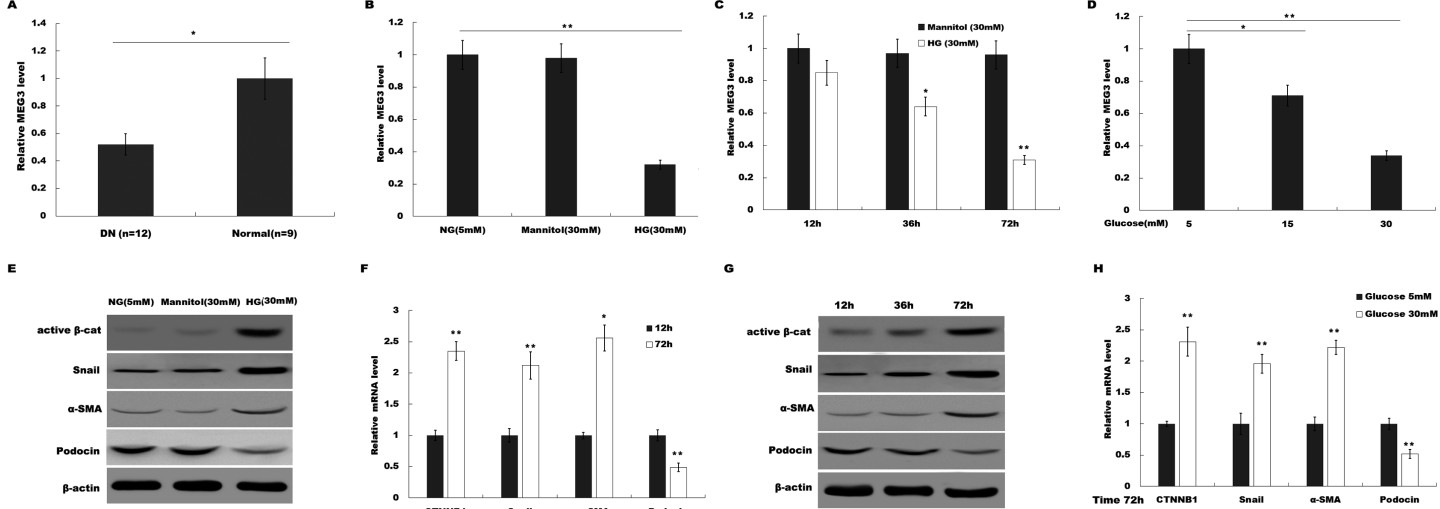

**Figure 1** **HG suppressed MEG3 expression in podocytes.** (A) The expressions levels of MEG3 in DN renal tissues ($n = 12$) and healthy controls ($n = 9$) are assessed using qPCR. (B) The expressions levels of MEG3 are assessed using qPCR in podocytes treated with normal glucose (5 mM), high glucose (HG, 30 mM) or Mannitol. (C–D) qPCR analysis showed the level of MEG3 in various conditions as indicated. The results from qPCR showed that MEG3 was decreased in a (C) time and (D) dose-dependent manner. (E) Cell lysates were immunoblotted with specific antibodies against active β-catenin (Active β-cat), snail1, α-smooth muscle actin (α-SMA), podocin and β-actin. Representative western blotting showed the expression of β-catenin target genes in various conditions as indicated. (F–H) qPCR and western blot analyses showed the expression level of podocin and β-catenin target genes in a time- and dose-dependent manner. *$P < 0.05$; **$P < 0.01$. Active β-cat; catenin beta-1 (CTNNB1); HG, high glucose.

## Statistical analysis

The SPSS 16.0 statistical analysis software (IBM Corporation, Armonk, NY, USA) was used for data analysis. The results were presented as mean ± standard deviation from three or more independent experiments. The difference between two groups was compared using two-tailed student's *t*-test, or one-way analysis of variance followed by the Scheffé test. A value of $P < 0.05$ was considered statistically significant.

## RESULTS

### MEG3 expression was decreased in podocytes after HG treatment

To investigate the possible role of MEG3 in HG-induced podocyte injury, we firstly assessed the level of MEG3 in renal tissues of DN patients and HG (30 mM)-treated podocytes using qPCR analysis. As shown in Fig. 1A, MEG3 expression levels were significantly reduced in renal tissues of DN patients compared to normal renal tissues. Figure 1B showed that HG treatment resulted in a decrease of MEG3 expression. We further investigated the effect of glucose concentration and treatment time on MEG3 expression. Figures 1C and 1D showed that MEG3 expression levels were significantly decreased in a time-dependent (Fig. 1C) and dose-dependent (Fig. 1D) manner. In addition, HG stimulation activated β-catenin and upregulated the expression of its target genes (a-SMA and snail1) in podocytes (Fig. 1E). HG also inhibited the expression of podocin (Fig. 1E), a podocyte-specific marker, indicating that HG treatment induced podocyte injury. HG increased the a-SMA and snail1 expression and reduced the podocin expression in a time-dependent and dose-dependent manner (Figs. 1F and 1G). These data

indicate the correlation between MEG3 and Wnt/β-catenin-mediated podocyte injury following HG treatment.

## MEG3 overexpression attenuated HG-induced podocyte injury

We then investigated the role of MEG3 in podocyte injury during HG treatment. pcDNA3-MEG3 was constructed and transfected into podocytes to overexpress MEG3. Figure 2A showed that pcDNA3-MEG3 transfection effectively enhanced MEG3 expression in podocytes. Functional studies demonstrated that HG activated β-catenin, upregulated the expression of a-SMA and snail1 in podocytes, whereas MEG3 overexpression reversed these effects (Fig. 2B). The HG-induced decrease in podocin expression was reversely enhanced with MEG3 overexpression (Fig. 2B). The cell viability was assessed by CCK-8 assay in HG-treated podocytes in the presence or absence of MEG3 overexpression. Figure 2C showed that overexpression of MEG3 alleviated HG-induced decrease of cell viability in podocytes. In addition, MEG3 overexpression repressed HG-induced enhancement of intracellular ROS level in podocytes (Fig. 2D). The cell migration determined by transwell assay showed that MEG3 overexpression markedly inhibited the migration ability of the podocytes in the presence of HG (Figs. 2E and 2F). These data demonstrated that MEG3 possessed the capacity to repress HG-induced increase of migration ability and ROS production and decrease of cell viability in podocytes.

## MEG3 improved renal function in diabetic rats

We then investigated the role of MEG3 in regulating renal function in diabetic rats. Rat models of DM were established by intraperitoneal injection of STZ and pcDNA3-MEG3 was administered to diabetic rats and the blood glucose level remained above 16.7 mmol/L after MEG3 overexpression in vivo. Figure 3A showed that MEG3 levels were decreased in renal tissues of diabetic rats compared with control, while pcDNA3-MEG3 treatment enhanced MEG3 expression. In addition, compared with normal control rats, overexpression of MEG3 significantly improved kidney function as demonstrated by decreased level of urine albumin to creatinine ratio and urine albumin excretion rate (Table 1). The results from qPCR showed that the expression of β-catenin, snail1 and α-SMA was increased and the expression of podocin was decreased in diabetic rat and mostly restored to control rats' level after overexpression with MEG3 (Fig. 3B). IHC staining also revealed that the increase of β-catenin, snail1 expression and the decrease of podocyte-specific marker synaptopodin in diabetic rat were reversely regulated with MEG3 overexpression (Figs. 3C and 3D). These results further demonstrated the protective role of MEG3 on renal function in diabetic rats.

## MEG3 protected against podocyte injury by inactivating Wnt/β-catenin signaling

MEG3 inhibits podocyte injury and suppresses Wnt/β-catenin signaling activation in DN. Given the important role of Wnt/β-catenin signaling in the pathogenesis of podocyte injury, we thus speculated whether MEG3 protects against podocyte injury by inactivating
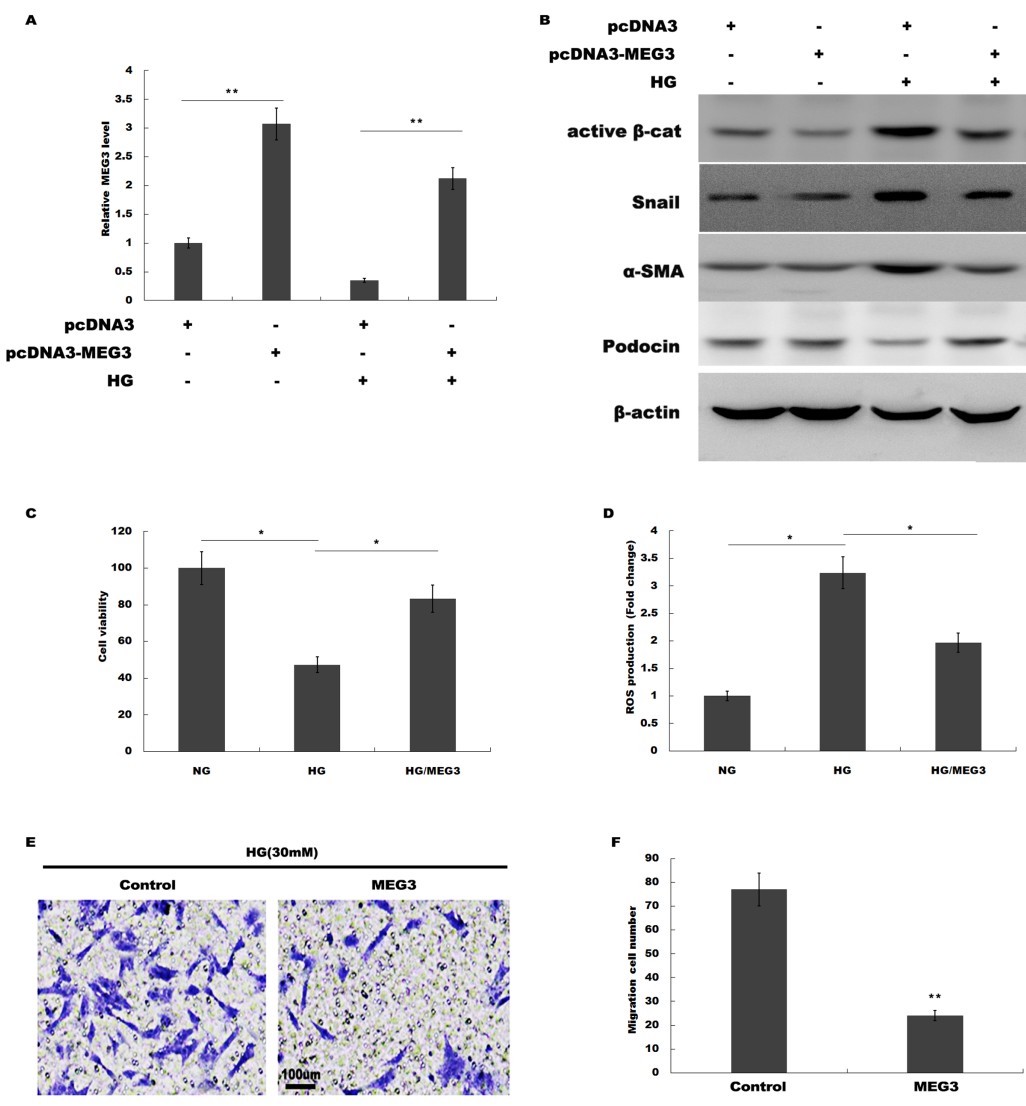

**Figure 2 MEG3 overexpression alleviated HG-induced podocyte injury.** (A) qPCR analysis showed that pcDNA3-MEG3 increased MEG3 expression in HG cultured podocytes. (B) Representative western blotting showed the expression of Active β-cat and its target genes in various conditions as indicated. (C) Cell viability was assessed using CCK-8 assay kit in podocytes treated with HG with or without MEG3 overexpression. (D) reactive oxygen species (ROS) production was assessed using ELISA assay kit in podocytes treated with HG with or without MEG3 overexpression. (E and F) Transwell migration assay and quantitative data showed a decreased migration of overexpression MEG3 in cultured podocytes. Scale bar, 100 μm. *$P < 0.05$; **$P < 0.01$. Active β-cat, active β-catenin.

Wnt/β-catenin. To prove it, podocytes were treated with HG in the presence or absence of MEG3 overexpression and the activation of Wnt/β-catenin signaling was assessed in vitro. The TOPFlash/FOPFlash luciferase reporter system was first used to directly assess the activation of β-catenin. The luciferase reporter map is listed in Fig. 4. Figure 5A showed that the relative luciferase activity was enhanced in HG-treated podocytes, whereas MEG3 overexpression markedly repressed luciferase activity. Western blot analysis further

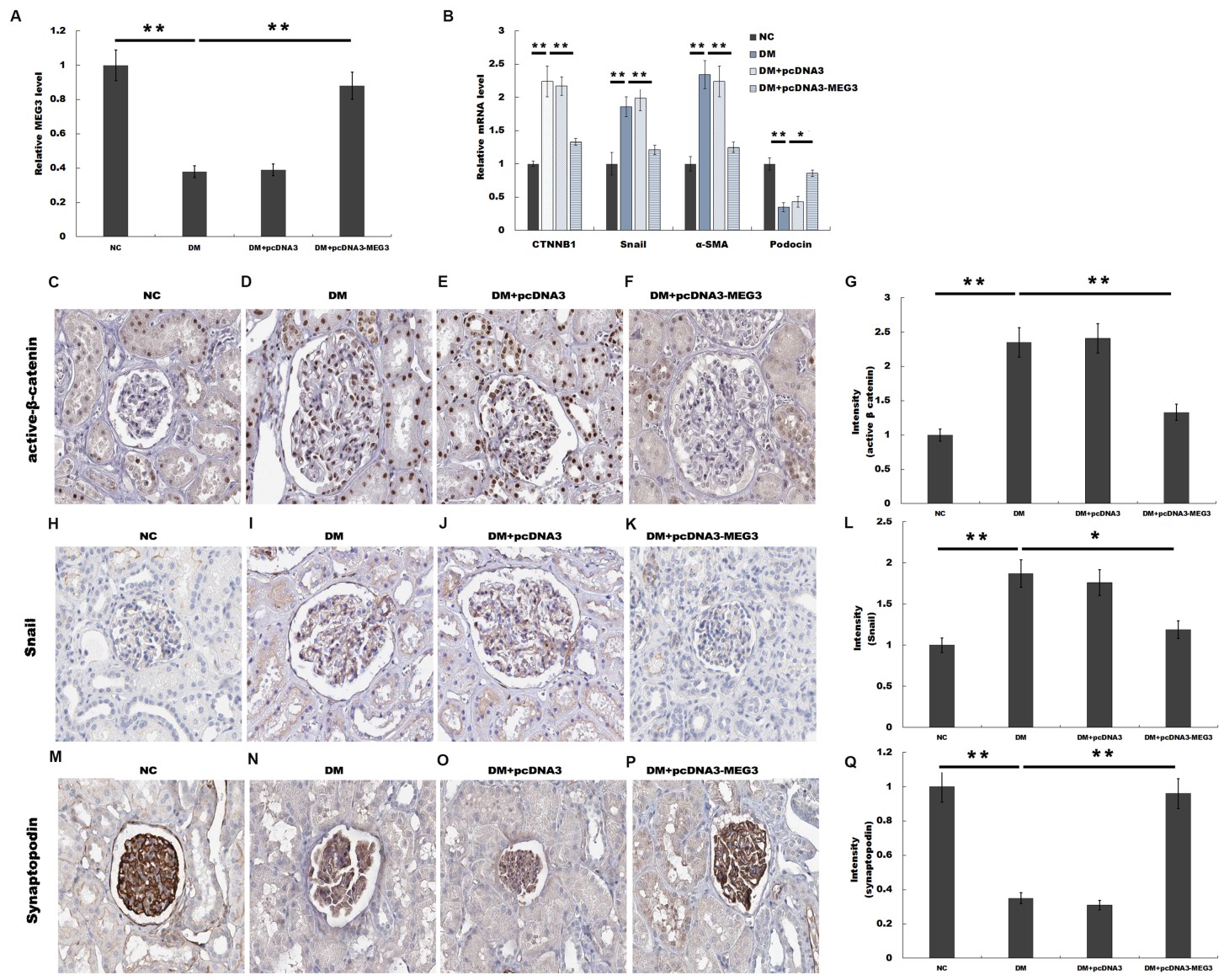

**Figure 3 MEG3 improved renal function in diabetic rats.** The mRNA level of MEG3 in renal tissues in diabetic rats (A) and CTNNB1, snail1, α-SMA, and podocin in Glomeruli (B) in various groups of diabetic rats as indicated. (C–Q) Representative immunohistochemical staining and quantitative data showed the expression of β-catenin and target genes in various groups as indicated. Frozen rat kidney paraffin-embedded sections were immunostained for Active β-cat, snail1 and synaptopodin. *$P < 0.05$; **$P < 0.01$. Active β-cat, active β-catenin; CTNNB1, catenin beta-1; HG, high glucose.

**Table 1 Parameters for diabetic rats treated with pcDNA3-MEG3 at week 12.** DR-pcDNA3, diabetic rats treated with *m* pcDNA3 control; DR-pcDNA3-MEG3 diabetic rats treated with *m* pcDNA3-MEG3; urine albumin to creatinine ratio; urine albumin excretion rate.

| Variables | DR-pcDNA3 | DR-pcDNA3-MEG3 |
|---|---|---|
| UAER | 1.73 ± 0.44 | 1.02 ± 0.11** |
| UACR | 24.39 ± 2.21 | 15.98 ± 1.12** |

**Note:**
** $P < 0.01$.

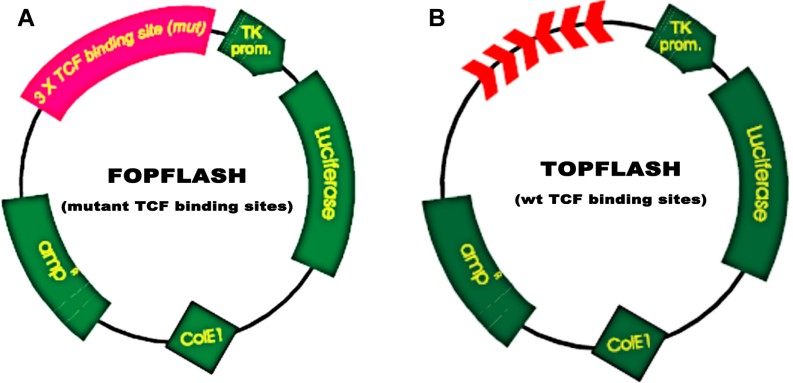

**Figure 4 TOPFlash/FOPFlash luciferase reporter map.** (A) TOPFLASH. Transfection grade T-cell factor (TCF) reporter plasmid containing two sets (with the second set in the reverse orientation) of three copies of the TCF binding site (wild type) upstream of the Thymidine Kinase (TK) minimal promoter and Luciferase open reading frame. (B) FOPFLASH. Transfection grade TCF reporter plasmid containing three copies of the TCF binding site (mutated) upstream of the TK minimal promoter and Luciferase open reading frame. This plasmid serves as a negative control TOPFLASH which contains wild type TCF binding sites.

verified that MEG3 inhibited HG-induced nuclear translocation of β-catenin, indicating the inhibitory effect of MEG3 on Wnt/β-catenin activation (Fig. 5B). After activation of Wnt, β-catenin translocates to nucleus and promotes the transcription of downstream target genes. The expression of target genes a-SMA and snail1 was then assessed after treatment with HG in the presence or absence of MEG3 overexpression. As shown in Fig. 5C, a-SMA and snail1 expression level was increased in HG-treated podocytes, whereas MEG3 overexpression significantly inhibited HG-induced upregulation of a-SMA and snail1.

Finally, we investigated whether MEG3 protected against podocyte injury by inactivating Wnt/β-catenin signaling. As expected, western blot analysis showed that MEG3 upregulated the expression of podocin in the presence of HG, whereas Wnt activation induced by SKL2001 partially destroyed the protective effect of MEG3 (Fig. 6A). More important, the results from CCK-8, transwell and ROS production assay showed that MEG3 suppressed HG-induced decrease of cells viability and increase of ROS production and cell migration in podocytes, whereas SKL2001 treatment partially destroyed the protective effect of MEG3 (Figs. 6B–6E). Taken together, these results demonstrated that MEG3 protects against podocyte injury via inactivating Wnt/β-catenin signaling in DN.

## DISCUSSION

In the current study, the roles of MEG3 in protecting against podocyte injury and potential molecular mechanism were investigated. Our data demonstrate that (i) MEG3 expression was decreased after HG treatment in podocytes, (ii) MEG3 overexpression attenuated HG-induced podocyte injury, (iii) MEG3 contributed to improve renal function in diabetic rats, (iv) MEG3 protected against podocyte injury by inactivating Wnt/β-catenin signaling. These data identified the important role and underlying mechanism of MEG3

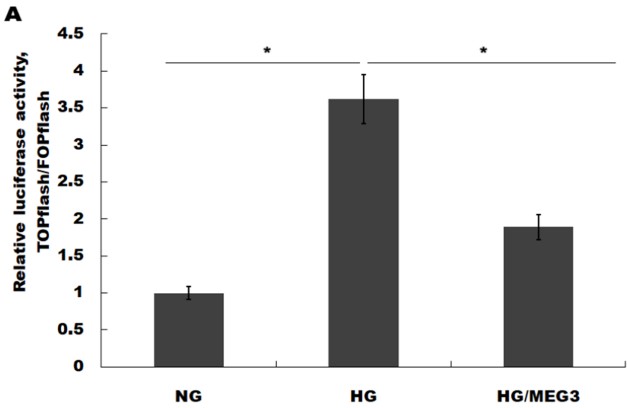

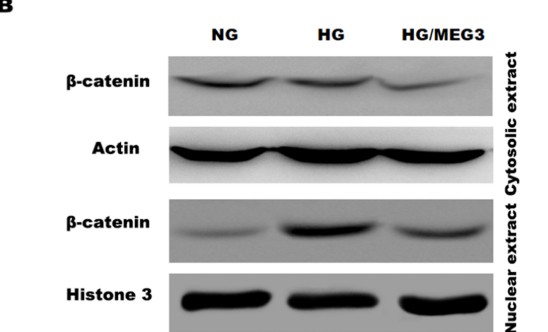

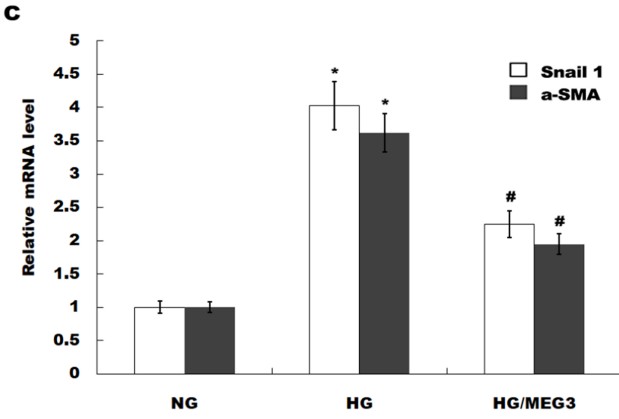

**Figure 5 MEG3 inhibited the activation of Wnt/β-catenin signaling in podocytes.** (A) The activation of canonical Wnt/β-catenin signaling was assessed in podocytes by TOPflash/FOPflash after MEG3 overexpression. $*P < 0.05$. (B) Western blot analysis showed that the nuclear translocation of β-catenin induced by HG was inhibited by MEG3 overexpression. (C) The relative snail1 and a-SMA mRNA expression was analyzed using qPCR in podocytes after MEG3 overexpression in the presence of HG. $*P < 0.05$ vs. NG group, $\#P < 0.05$ vs. HG group.

in inhibiting podocyte injury in DN and may provide a novel opportunity to the therapy of DN.

As a frequent microvascular complication of DM, specific clinical hallmarks of DN have been revealed including the recession of glomerular filtration rate and the progressive

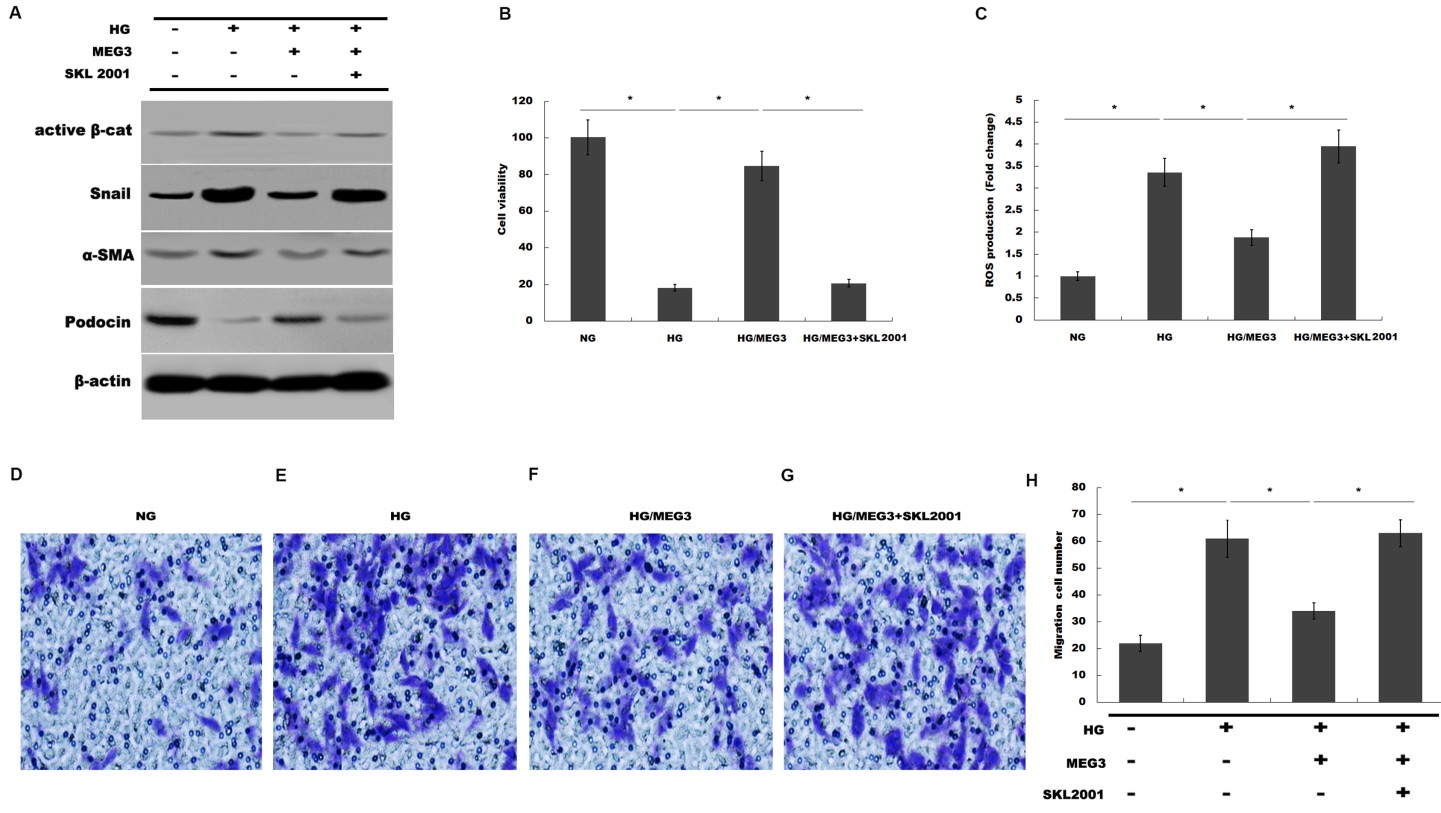

**Figure 6 MEG3 inhibited podocyte injury by inhibiting Wnt/β-catenin signaling.** (A) The protein levels of Active β-cat, snail1, a-SMA and podocin in podocyte were assessed using western blot analysis in podocytes. (B) Cell viability was assessed using CCK-8 assay kit in podocytes treated with indicated reagents. (C) ROS production was assessed using ELISA assay kit in podocytes treated with indicated reagents. (D–H) Transwell assay and quantitative analysis for cell migration in podocytes treated with indicated reagents. *$P < 0.05$.

urinary albumin excretion, eventually developing end-stage kidney diseases (*Giunti, Barit & Cooper, 2006*). Podocytes are an important ingredient of glomerular filtration barrier which possesses special biological and physiological action to maintain the structure and function of kidneys (*Teng, Lukasz & Schiffer, 2012*). The loss or injury of podocytes is the crucial cause of proteinuria (*Yuan et al., 2017*). Up to date, the molecular mechanism of podocyte injury has been identified gradually in which lncRNAs are considered to take major part in regulation of podocytes function (*Long & Danesh, 2018*). *Hu et al. (2017)* reported that lncRNA-MALAT1 expression is enhanced in renal tissues of STZ-induced DN compared with normal renal tissues. HG treatment also results in the increase of MALAT1 expression in podocytes. They further demonstrated that knockdown of MALAT1 inactivates Wnt/β-catenin signaling and protects against HG-induced podocyte injury. More important, MALTA1 inhibition restores podocytes function by the mediation of Wnt/β-catenin signaling (*Hu et al., 2017*). Given the crucial role of Wnt/β-catenin signaling in the development and progression of DN, MALAT1 might be a novel biomarkers and therapeutic targets involved in DN.

Our previous studies identified MEG3 as a tumor suppressor gene in bladder cancer (*Ying et al., 2013*). As with other studies, our work demonstrated that MEG3 represses

cancer cells proliferation by inactivating Wnt signaling pathway (*Deng et al., 2018*; *Li et al., 2018*; *Ying et al., 2013*). However, the role of MEG3 in podocyte injury remains unknown. We thus investigated the interaction of MEG3 with Wnt signaling and whether the interaction is correlated with podocyte injury. Current data showed that MEG3 levels are reduced in renal tissues of DN patients compared to normal renal tissues. HG treatment also leads to the decreased expression of MEG3 in a time-dependent and dose-dependent manner. Then we investigate the function of MEG3 on podocyte injury in the presence of HG. Functional studies showed that HG represses podocin and synaptopodin expression in vitro and in vivo, whereas MEG3 overexpression restores their levels, indicating the protective role of MEG3 in podocyte injury. Indeed, overexpression of MEG3 alleviates HG-induced decrease of cell viability, represses HG-induced enhancement of intracellular ROS level and inhibits the migration ability of podocytes in the presence of HG. We further demonstrated the protective role of MEG3 on renal function in diabetic rats in vivo. As expected, HG treatment results in the activation of Wnt/β-catenin signaling, whereas MEG3 overexpression inhibits HE-induced activation of Wnt/β-catenin signaling. Finally, we found that MEG3 protects against podocyte injury by inactivating Wnt/β-catenin signaling.

## CONCLUSION

The interplay between MEG3 and Wnt/β-catenin signaling affects the onset of podocyte injury and renal dysfunction. Most importantly, MEG3 overexpression can interrupt the activation of Wnt/β-catenin signaling and restore the renal dysfunction with protection from podocyte injury. Therefore, our finding provides a new insight on potential drug target to treat podocyte injury and DN.

### Funding
The authors received no funding for this work.

### Competing Interests
The authors declare that they have no competing interests.

### Author Contributions
- Xiajing Che conceived and designed the experiments, prepared figures and/or tables, authored or reviewed drafts of the paper, approved the final draft.
- Xin Deng performed the experiments, prepared figures and/or tables, authored or reviewed drafts of the paper, approved the final draft.
- Kewei Xie performed the experiments, prepared figures and/or tables, authored or reviewed drafts of the paper, approved the final draft.
- Qin Wang analyzed the data, prepared figures and/or tables, authored or reviewed drafts of the paper, approved the final draft.
- Jiayi Yan analyzed the data, prepared figures and/or tables, authored or reviewed drafts of the paper, approved the final draft.

- Xinghua Shao conceived and designed the experiments, performed the experiments, analyzed the data, contributed reagents/materials/analysis tools, prepared figures and/or tables, authored or reviewed drafts of the paper, approved the final draft.
- Zhaohui Ni conceived and designed the experiments, performed the experiments, analyzed the data, authored or reviewed drafts of the paper, approved the final draft.
- Liang Ying conceived and designed the experiments, prepared figures and/or tables, authored or reviewed drafts of the paper, approved the final draft.

## Animal Ethics

The following information was supplied relating to ethical approvals (i.e., approving body and any reference numbers):

The Animal Ethics Committee of Shanghai Jiaotong University School of Medicine provided full approval for this research (SYKX-2008-0050).

## Data Availability

The raw measurements are available in the Supplemental Files.

## Supplemental Information

Supplemental information for this article can be found online at http://dx.doi.org/10.7717/peerj.8016#supplemental-information.

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
