# Peer review of "Long noncoding RNA MEG3 suppresses podocyte injury in diabetic nephropathy by inactivating Wnt/β-catenin signaling"

_PeerJ, doi:10.7717/peerj.8016_

## Round 0.1 · original submission · Major Revisions

The reviewers have identified major concerns related to methodology; specifically, overexpression of MEG3 in vivo using pcDNA3 plasmid has to be reconfirmed with additional data.

Reviewer 1 ·

Basic reporting

1.overall writing is poor and needs significant improvements
2. Literature review- The authors did not have enough background literature search and I was bit lost as to why they choose to study MEG3 Lnc out of nowhere. They need to do more literature search and make it clear in the introduction as to why they choose MEG3. It was kind of choppy, they explained MALATI Lnc and jumped to MEG3
3. article structure- They have done good job of data presentation

Experimental design

1. The study conducted is original and fits the Aims and Scope of journal
2. They need to emphasize Why the whole project was undertaken, research question is poorly defined.
3. The research performed is rigorous, couple of issues need to be addressed as explained in comments to the author
4. Materials and Methods need to be more specific there were instances where I was lost as to how did they do the experiment. A. cell counting protocol- they do not mention how they differentiated live and dead cells. B. ROS production-I am confused what kit they used.

Validity of the findings

1.The study is well designed, data is presented well and the reader understands to logical flow
2. The study need to be more robust. The authors assume that when plasmid is injected i it is taken up only by podocytes- it can be taken up by any tissue and observed changes could be result of changes in general metabolism.
3. Conclusions are well stated and data supports cause and effect in-vitro. However in-vivo data needs more rigorous testing.

Additional comments

Major comments.
1. writing needs significant improvement. The reader is often confused and lost because of the poor language usage. Logical connection, cohesion and flow is lacking. Please do extensive literature review and rewrite the introduction as to why LnC MEG3 study was undertaken in DN.
2. Materials and Methods section writing needs significant improvement- please write in brief the principles involved for all the methods. For example ROS was detected by ELISA- I am confused they detected ROS or ROS modified proteins?
3. In Figure 1C- DMSO was used - please explain why this was used
4.They used pCDNA 3 vector- This could be taken up by any tissue in the body and all other cell types in the kidney. What makes the authors believe that effects observed are specifically because of podocytes in the kidney . could you please give more evidence to show MEG3 is expressed only in podocytes ?.
5. The authors should also carry out blood glucose and other parameters after administering MEG3 to rule out MEG3 Lnc altering the metabolism.
6. In Diabetic Neprhopathy almost all cell types are affected. can the authors give more evidence to rule out changes in other cell types in the kidney after injecting MEG3 Lnc?

Reviewer 2 ·

Basic reporting

No comments.

Experimental design

In the current study, Xiajing Che and colleagues have explored the role of Long noncoding RNA MEG3 in diabetic nephropathy. The methods have not been sufficiently described. In particular, how the in vivo studies were carried out specifically in Figure-3 has not been described. How was MEG3 overexpressed in podocytes in vivo using a plasmid?

Validity of the findings

Although the in vitro studies with high glucose treatment and expression studies in patients and animal tissues are novel. The functional studies have not been described well. The studies described in Figure-3 are particularly troublesome. How were the authors able to overexpress MEG3 in podocytes in vivo using a plasmid? This method is not described in the methods sections. To be able to overexpress a gene in vivo using pcDNA3 plasmid is currently technically impossible. These results raise serious questions on how the experiments were carried out and the data interpretation.

Additional comments

The data presented in Figure-3 raises critical questions on the validity of this study.

---

## Round 0.2 · accepted · Accept

Please coordinate with PeerJ staff to include the following before production: Please include luciferase reporter map

Reviewer 1 ·

Basic reporting

no comment

Experimental design

no comment

Validity of the findings

no comment

Additional comments

Please include luciferase reporter map
double check ROS detection method- ELISA kit- I am confused what did you detcet?

Reviewer 2 ·

Basic reporting

No Comment

Experimental design

No Comment

Validity of the findings

No Comment

Additional comments

The authors have addressed all the concerns/comments adequately.